# Effect of Freeze–Thaw Cycles on the Freshness of Prepackaged *Penaeus vannamei*

**DOI:** 10.3390/foods13020305

**Published:** 2024-01-18

**Authors:** Shouchun Liu, Luyao Zhang, Jing Chen, Zhuyi Li, Meijiao Liu, Pengzhi Hong, Saiyi Zhong, Haifeng Li

**Affiliations:** 1College of Food Science and Technology, Guangdong Ocean University; Guangdong Provincial Key Laboratory of Aquatic Product Processing and Safety; Guangdong Provincial Engineering Technology Research Center of Seafood, Guangdong Provincial Engineering Technology Research Center of Aquatic Prepared Food Processing and Quality Control; Guangdong Modern Agricultural Science and Technology Innovation Center, Zhanjiang 524088, China; liusc_zjwlab@163.com (S.L.); 15292604455@163.com (J.C.); s2416001752@163.com (Z.L.); lmj2504@163.com (M.L.); hongpengzhi@126.com (P.H.); 13692794834@163.com (H.L.); 2Southern Marine Science and Engineering Guangdong Laboratory, Zhanjiang 524004, China; 15646705235@163.com

**Keywords:** *Penaeus vannamei*, temperature fluctuation, quality characteristics, moisture change, electronic nose sensor, shelf life

## Abstract

The effect of temperature fluctuations on the freshness of shrimp in simulated trays was investigated by setting a freeze–thaw (F-T) cycle of 12 h after freezing at −20 °C and thawing at 1 °C under refrigeration. The results showed that the shrimp’s physicochemical properties deteriorated to different extents with the increase in F-T cycles. The total colony count of shrimp was 6.07 lg CFU/g after 21 cycles, and the volatile saline nitrogen content reached 30.36 mg/100 g, which exceeded the edible standard. In addition, the sensory quality and textural properties (hardness, elasticity, chewiness, and adhesion) declined to different degrees with increased F-T cycles. LF-NMR and protein property measurements showed that F-T cycles resulted in reduced water holding capacity and protein denaturation, which were the main factors leading to the deterioration of shrimp quality. Furthermore, flavor changes were analyzed using an electronic nose sensor to establish a freshness model. The W1W, W1S, W2S, and W5S sensors were correlated with the quality changes in shrimp and used as the main sensors for detecting the freshness of *Penaeus vannamei*. As a result, to better maintain the overall freshness, temperature fluctuations should be minimized in sales and storage, and fewer than 8 F-T cycles should be performed.

## 1. Introduction

*Penaeus vannamei* is among the most cultivated shrimp worldwide, with fast growth, disease resistance, ease of culture, etc. It is an excellent desalination culture shrimp. *Penaeus vannamei* is rich in nutrients, protein, polyunsaturated fatty acids, and mineral elements [1], is flavorful and delicious, and is loved by consumers. However, shrimp quickly change during storage and become corrupted, which seriously affects their commercial value. For example, the shrimp body polyphenol oxidase and tyrosine oxidation reaction promote the generation of black change, affecting consumer acceptance [2]. Low-temperature storage, modified atmosphere packaging [3], additive packaging [4], and irradiation treatment methods [5] are currently used for the preservation of *Penaeus vannamei*.

Among them, cryopreservation (<−18 °C) effectively inhibits microbial growth and reproduction and enzyme activity, maintaining quality and extending shelf life [6]; this is the main method used for fresh storage [7]. However, the muscle quality of shrimp under frozen storage conditions still deteriorates, with texture changes, protein denaturation, a loss of juices [8], among other changes [9]. In particular, in the low-temperature transportation and circulation, storage, and consumption process at temperatures less than −18 °C (GB/T 31080-2014) [10], the inevitable temperature fluctuations will subject the shrimp body to a cyclic freeze–thaw (F-T) phenomenon [11], exacerbated by changes in its physicochemical properties and microbial spoilage, thus affecting the quality of *Penaeus vannamei* [12]. For example, the local thawing, refreezing, and recrystallization of shrimp muscle often occur during cold chain distribution storage, leading to denaturation, the growth of ice crystals in the muscle tissue, and the degradation of the structural properties of shrimp meat [13]. Dang et al. [14] found that temperature fluctuations accelerated the rate of deterioration of shrimp quality. Therefore, changes in shrimp quality and freshness due to F-T cycles are still a matter of concern.

In recent years, F-T cycle effects on product muscle moisture and quality caused by temperature fluctuations have received extensive attention. Repeated F-T cycles affect water distribution inside the food and form many ice crystals, most of which are big in size and growth in every other F-T cycle [15]. These crystals cause irreversible damage to the cells, including tissue structure disruption, lipid oxidation, and protein denaturation, thus decreasing muscle quality. Previous studies demonstrated that repeated F-T cycles affect the nutritional quality and consumer acceptance of seafood [16]. For example, Cheng et al. [17] found that repeated F-T cycles (−18 °C~4 °C) disrupted the beef muscle fiber network structure and reduced the quality and characteristics of beef, such as water-holding capacity, textural properties, and freshness. Therefore, temperature fluctuations should be minimized during storage. Boonsumrej et al. [18] showed that with the growing number of F-T cycles (−20 °C~0 °C), the thiobarbituric acid value and shrimp shear force increased, the muscle proteins’ salt-soluble nature decreased, and the muscle fibers bent and fractured. As a result, the quality of shrimp meat declined to various degrees. Sriket et al. [19] found that repeated F-T cycles (−20 °C~0–2 °C) resulted in the denaturation of shrimp meat proteins and decreased Ca^2+^-ATPase activity in black and white tiger shrimp, leading to myofibrillar fracture and the disorganization of the muscle structure. Thanonkaew et al. [20] also reported that temperature changes (−18 °C~0–2 °C) during F-T cycling accelerated cuttlefish fat oxidation. Repeated freezing–thawing caused the rupture of cuttlefish myocytes and the release of cytosolic enzymes into the sarcoplasm, promoting fat oxidation. In summary, research on F-T cycles in seafood has been thorough, but the effects of F-T cycles on the freshness of prepackaged *Penaeus vannamei* should be further studied.

Temperature fluctuations are unavoidable during the cold chain transportation, storage, and consumption of aquatic products. Unsold shrimp in the market are frozen and resold, leading to multiple F-T cycles. Therefore, this experiment aimed at studying the changes in the freshness of *Penaeus vannamei* after F-T cycles. The odor characteristics and freshness changes were analyzed using sensory score, texture, pH value, total volatile basic nitrogen (TVB-N), total plate count (TPC), and electronic nose technology, and a rapid detection model of shrimp freshness was established. A comprehensive analysis of the changes in shrimp freshness after F-T cycles provides some theoretical guidance for maintaining the quality of *Penaeus vannamei* in the sales process.

## 2. Materials and Methods

### 2.1. Materials

Fresh *Penaeus vannamei* (width of approximately 18–23 mm) were purchased from Dongfeng Market in Xiashan District (Zhanjiang, China). Plate counting agar (022070) was purchased from Huankai Biotechnology Co., Ltd. (Shaoguan, China); sodium chloride (7647-14-5), anhydrous sodium carbonate (497-19-8), boric acid (10043-35-3), and sodium hydroxide (1310-73-2) were purchased from Xilong Science Co., Ltd. (Shenzhen, China); hydrochloric acid titration standard solution (7647-01-0) was purchased from Anpu Experimental Technology Co., Ltd. (Shanghai, China); anhydrous potassium carbonate (584-08-7) and gum arabic (9000-01-5) were purchased from Aladdin Reagents Co., Ltd. (Shanghai, China).

### 2.2. Sample Preparation

The uniform size and fresh *Penaeus vannamei* were selected and killed by breaking ice, cleaning, and decontaminating. The whole shrimp was obtained after the head, shell, tail, and thread were removed; then, the water was drained and set aside. The treated shrimps were placed in trays and covered with fresh-keeping film. The F-T temperature and time were as follows: −20 °C frozen 12 h (equivalent to the night frozen storage), and then 1 °C refrigerated conditions thawed 12 h (equal to the daytime refrigerated sales); the whole process is an F-T cycle. The shrimp refrigerated at 1 °C were used as the control group to determine the freshness indices.

### 2.3. Determination of Physicochemical Properties

#### 2.3.1. Sensory Evaluation

According to a sensory evaluation by Rahman et al. [21], 6–8 sensory evaluators (aged 20–30 years) with evaluation experience were selected to evaluate the odor, texture, appearance, and boiled quality of each group of samples. The average sensory score was combined with the overall sensory evaluation for analysis. The specific scoring criteria are shown in Appendix A.

#### 2.3.2. Total Plate Count (TPC)

TPC was determined through the plate counting method according to the Chinese national standard GB 4789.2-2022 [22]. After grinding, shrimp samples (25.00 g) were homogenized in a sterilized centrifuge tube containing 225 mL of saline. Serial dilutions were performed, and 3 suitable dilutions were selected. A total of 1 mL of the dilution was taken and poured into the plate agar, rotating the Petri dish to form a homogeneous mixture. After the agar solidified, it was turned over and incubated at 30 °C ± 1 °C for 72 h ± 3 h, expressed as CFU/g.

#### 2.3.3. pH and Total Volatile Basic Nitrogen (TVB-N) Measurement

A 3.0 g minced *Penaeus vannamei* sample was mixed with 25 mL distilled water and left for 30 min. The pH value of the collected filtrate was determined using the PHS-25 pH meter (INESA Scientific Instrument Co., Ltd., Shanghai, China).

According to the Chinese national standard GB 5009.228-2016 [23], the TVBN content in the samples was determined using the microdiffusion method. A total of 1 mL boric acid indicator was added to the inner dish of the diffusion dish, and 1 mL of filtered shrimp juice and 1 mL of saturated potassium carbonate solution were added to the outer dish, which was mixed thoroughly and left in a volatile state. After standing for 24 h, the samples were titrated with titrated hydrochloric acid (0.01 mol/L HCl). A control test was also performed with distilled water as a blank group.

#### 2.3.4. Texture Property Analysis (TPA)

The thawed shrimps were equilibrated at room temperature for 30 min, and the second and third sections of the shrimp chest were collected. The direction of the downward pressure of the texture meter (TA.XTplusC, Stable Micro Systems Ltd., Surrey, UK) was perpendicular to the direction of the muscle fibers of the shrimp; then, the shrimp was analyzed using the texture meter. The parameters were set as follows: a pretest speed of 3.00 mm/s, a test speed of 1.00 mm/s, a posttest speed of 1.00 mm/s, a downward distance of 50%, a test time of 5 s, a trigger force of 10 g, and probe Model P5.

### 2.4. Extraction of Myofibrillar Protein (MP)

The method used to prepare MP followed a method by Peng et al. [24] with slightly modifications. A tenfold volume of PBS buffer (pH 7.0) was added to the shrimp meat, homogenized at 8000 r/min for 1 min, and centrifuged at 8000 r/min for 20 min at 4 °C; the precipitate was retained and washed twice. Then, the precipitate was mixed with a threefold volume of PBS buffer (containing 0.6 mol/L KCl, pH 7.0). The mixture was homogenized for 1 min, left at 4 °C for 1 h, and then centrifuged at 10,000 r/min for 15 min. The supernatant obtained was MP solution.

### 2.5. Sodium Dodecyl Sulfate-Polyacrylamide Gel Electrophoresis (SDS-PAGE)

SDS-PAGE was performed with reference to Lan et al. [25] with minor modifications. The concentration of MP was determined by biuret, and the concentration of MP was adjusted to 1 mg/mL. The MP solution was mixed with SDS-PAGE sample buffer (2×), heated in boiling water for 5 min, and loaded into the gel. The electrophoresis program was set to 80 V for 30 min, and then 100 V for 80 min. After dyeing and bleaching, the gel was photographed on a gel imager.

### 2.6. Moisture Measurement

#### 2.6.1. Moisture Content

The samples’ moisture content was determined using a Halogen Lamp Moisture Tester (HX204, METTLER TOLEDO, Zurich, Switzerland). A certain amount of crushed shrimp samples was weighed and spread flat on a tin foil disk, and then placed into the instrument for testing; each set of experiments was repeated three times in parallel.

#### 2.6.2. Water Holding Capacity (WHC)

WHC was determined through centrifugation [17,26]. Shrimp meat of weight m_1_ was placed in a centrifuge tube with filter paper at the bottom and then placed in a freezing centrifuge and centrifuged at 3100× *g* for 5 min at 4 °C (to prevent the denaturation of the proteins). After centrifugation, the shrimp meat m_2_ was weighed again. WHC was calculated according to the following formula:$$WHC(\%)=\frac{m_{2}}{m_{1}}\times 100\%$$

#### 2.6.3. LF-NMR and MRI

Referring to the method by Wang et al. [27], the nuclear magnetic resonance analysis (NMI20-025V-I, Niumag Electric Corporation, Shanghai, China) was used. The parameters were set as follows: magnet temperature 32 °C, sampling point 6160, τ120 μs, scanning number 6, and echo number 8000. The T_2_ value of the shrimp sample was generated using inversion software fitting.

The proton density imaging of the samples was determined through MRI, and the relevant parameters were set by Cheng et al. [28], with a repeated waiting time of 500 ms, and an echo time of 18.2 ms. The shelled shrimp, wrapped in cling film, was placed in a nuclear magnetic tube, and its signal-to-noise ratio and image clarity were adjusted to obtain an imaging map.

### 2.7. Electronic Nose

An electronic nose detection system (PEN3, AIRSENSE Analytics GmbH, Mecklenburg, Germany) consisting of 10 different metal sensors was used to determine the odor of *Penaeus vannamei*. The sensor array and performance of the electronic nose are specified in Appendix A. Samples (3.0 g) were weighed into a 100 mL beaker, wrapped with two layers of cling film, and equilibrated at room temperature for 30 min. The assay parameters were set according to the method of [29], as follows: sample preparation time = 5 s; sensor cleaning time = 60 s; injection volume = 800 mL/min; and assay time = 120 s. Each sample was measured 4 times in parallel.

### 2.8. Statistical Analysis

All experimental data were repeated three times and averaged, and the data were analyzed for significant differences (*p* < 0.05) using Duncan’s range of comparison test in SPSS 17.0 software. The electronic nose data were processed and analyzed with the Winster system, using principal component analysis (PCA), linear discriminant analysis (LDA), and loading analysis.

## 3. Results and Discussion

### 3.1. Sensory Evaluation

Sensory ratings are the most intuitive indicator of freshness and are essential to consumers’ buying decisions. The lower the sensory score is, the more serious the deterioration of food quality. Figure 1A shows that shrimp’s overall sensory quality changes under cold-storage and F-T cycles. With increasing storage time, the sensory quality of the samples decreased significantly (*p* < 0.05), such as the characteristic fresh and sweet smell was gradually replaced by ammonia and a fishy smell; the appearance changed from glossy silver-white to a yellow or reddened color without body surface gloss; and the shrimp surface changed from fresh to rotten. These changes may be due to the oxidation reaction between polyphenol oxidase and tyrosine, which produces a black color [30], gradually reducing the sensory color score. After 12 d of storage, the cold-storage group showed soft tissue, exhibited poor elasticity, and emitted strong ammonia and fishy odor, which was below the acceptable value. In contrast, the sensory quality of the F-T group samples decreased sharply after 16 days, and the pieces were inedible and more corrupt at 21 days. This result indicated that the F-T group’s threshold of acceptable sensory quality was 20 cycles.

### 3.2. Physicochemical Analysis

#### 3.2.1. TPC

TPC is an essential indicator for evaluating the food’s hygienic status, indicating the product’s degree of microbial contamination and whether it meets the national edible safety standards. Studies have shown that the total colony count of shrimp is less than 5.0 lg CFU/g for the first grade of freshness; 5.0 lg CFU/g ≤ total colony count < 6.0 lg CFU/g for the second grade of freshness; and if TPC is higher than 6.0 lg CFU/g, the shrimp is inedible [25], which can be used as a judgment criterion for the shelf life. Figure 1B shows the shrimp colony total changes under constant temperature refrigeration and F-T cycles. With the extension of storage time, the refrigeration group’s colony real growth rate was generally more significant than that of the F-T group (*p* < 0.05), reflecting that the freezing conditions can effectively inhibit the growth and reproduction of some microorganisms. The refrigeration group reached secondary freshness and tertiary freshness on the 6th and 12th day, respectively, i.e., the unacceptable value had been reached on the 12th day. In contrast, the TPC of the F-T group reached 5.18 lg CFU/g, i.e., secondary freshness, after 9 F-T cycles, and the TPC rose to 6.07 lg CFU/g, which was beyond the edible range, after 21 cycles. In summary, F-T inhibited some microorganisms’ physiological activity and reproduction rate and slowed down the degree of shrimp corruption, so the maximum number of F-T cycles was 20.

#### 3.2.2. TVB-N

The TVB-N content is the most effective technique used to assess shrimp freshness. TVB-N in seafood includes ammonia (NH_3_), dimethylamine (C_2_H_7_N), and trimethylamine (C_3_H_9_N) [31]. During storage, the content of TVB-N in seafood increases due to the production of amino acids through microbial decarboxylation in vivo [32,33]. Therefore, the freshness of shrimp can be evaluated using the TVB-N content. In general, a TVB-N value greater than 20 mg/100 g is the freshness threshold of *Penaeus vannamei*, 20–30 mg/100 g is the acceptable threshold of quality, and a value greater than 30 mg/100 g means that the shrimp are inedible [34,35]. As shown in Figure 1C, the TVB-N content of shrimp gradually increased with the extension of storage time. The main reason was that during storage, shrimp were slowly decomposed by microorganisms and endogenous enzymes, which generated amino acids and nitrogen-containing substances, such as ammonia and amines. Among them, the TVB-N content of the cold-storage group reached the secondary freshness threshold and the inedible threshold on the 4th day (18.58 mg/100 g) and the 8th day (30.43 mg/100 g), respectively, while the F-T group reached the acceptable quality threshold and the unacceptable freshness threshold on the 8th and 21st days, respectively. Therefore, the F-T environment could partially inhibit the increase in the content of TVB-N produced by microorganisms or enzymes, and its preservation effect was slightly better than that of the cold-storage group (*p* < 0.05).

#### 3.2.3. pH

Figure 1D shows the changes in the pH value of shrimp under refrigeration with constant temperature and under F-T conditions. During storage, the pH value of the shrimp tended to first decrease and then increase. This probably occurred because the glycogen of *Penaeus vannamei*, as a crustacean, was first decomposed in the early storage stage to produce acid substances, such as lactic acid, so that the pH value decreased temporarily. After that, with the continuous decomposition and depletion of glycogen, the protein in the body was continuously decomposed by microbial reproduction and endogenous enzymes to generate alkaline substances; therefore, the pH value reached the lowest point and then gradually increased [36]. During storage, the pH value of the cold-storage group began to improve on the third day and was consistently higher than that of the F-T group (*p* < 0.05). The pH of the F-T group showed a slow upward trend, probably because the F-T cycle process effectively inhibited the decomposition of proteins and other substances, reducing the alkaline substances. Sample corruption is greater at higher pH values, and the freshness of the shrimp also changes accordingly. Previous studies have found that the freshness of *Penaeus vannamei* is unacceptable at pH > 7.6 [37], so the cold-storage group and the F-T group reached the unacceptable threshold at 16 d and 23 d, respectively, and their quality was significantly reduced.

### 3.3. TPA

The storage of shrimp involves the following stages: rigidity, autolysis, and spoilage. Texture parameters can reflect the freshness of shrimp muscle. With the increase in storage time, the hardness, elasticity, chewiness, and cohesiveness of shrimp fluctuated; in particular, the elasticity index fluctuated wildly, which may result from the individual differences of shrimp. Nevertheless, the texture characteristics of shrimp showed a downward trend (Figure 2). The changes in the texture characteristics of fresh shrimp during cold storage may result from the weakening effect of endogenous and microbial enzymes on the proteolysis of myofibrillar protein and connective tissue [38]. Under the same storage time, the texture quality of shrimp in the F-T group decreased slowly compared to the cold-storage group, consistent with the above physicochemical properties. During cold storage, the protein density of shrimp muscle decreases. With the increase in storage time, the protein content in myofibrils may fall under the action of proteolytic enzymes, resulting in a decrease in the elasticity of shrimp muscle [39,40]. At the same time, some active microorganisms perform life activities in the shrimp, consume its water, and reduce the corresponding elastic texture. Therefore, water loss and protein denaturation during low-temperature storage are the main reasons for the quality changes in shrimp meat [41].

### 3.4. SDS-PAGE

SDS-PAGE was used to analyze the degradation of MP in shrimp under different storage conditions. As shown in Figure 3, the MP of shrimp consisted of myosin heavy chain (MHC, approximately 245 kDa), para myosin (PM, approximately 100 kDa), actin (approximately 45 kDa), and myosin light chain (MLC, about 18 kDa) [42]. At the beginning of storage, the MP sample was not significantly different from the control. With decreasing freshness, all MP bands became lighter to varying degrees. With repeated F-T cycles, the band intensities of MHC and actin gradually decreased, with a more rapid decline than that in the cold-storage group, especially during the spoilage period. This result was similar to the experimental results of Lan et al. [25], indicating that repeated F-T cycles exacerbated protein hydrolysis and oxidative denaturation, leading to the degradation or digestion of MPs and the breakdown of actin and MHC into smaller peptides [43]. In addition, the growth and recrystallization of extracellular ice crystals may concentrate solutes in the remaining liquid phase, leading to protein denaturation during F-T cycles [24,40]. This finding was consistent with the results of the above experiments that repeated F-T cycles led to MP degradation and reduced the quality of shrimp.

### 3.5. Moisture Variation

#### 3.5.1. Moisture Content and WHC

An essential factor that affects the flavor quality of aquatic products is their moisture content, which is related to the product’s color, taste, and tenderness and affects subsequent processing and storage. Figure 4A shows the change in moisture content under constant temperature refrigeration and F-T conditions. With increasing storage time, the moisture content of shrimp decreased continuously (79.90%→76.15%), similar to the moisture loss reported by others [44,45]. This may be due to the denaturation of shrimp muscle proteins and the damage of cell membrane structure under the action of endogenous enzymes or microbial reproduction, resulting in a decreased muscle water-holding capacity and moisture content [46]. Among them, the change in the moisture content of shrimp in the F-T group was relatively large, possibly due to the shrimp’s recrystallization after the F-T process. When part of the water in the shrimp crystallized, the solute concentration in the uncrystallized solution increased, so the enzyme activity in the body increased, the muscle protein denaturation was more apparent [47], and the muscle water-holding capacity was less than that of the cold-storage group (*p* < 0.05).

WHC refers to the ability of the shrimp to retain water under an external force, which directly reflects the strength of the shrimp’s ability to bind water [48]. A high WHC indicates that the internal structure of the shrimp is complex and compact, and the binding power of protein and water molecules is strong. With the decrease in freshness, the protein is decomposed and denatured, and the water-holding capacity of shrimp meat decreases accordingly [49]. As shown in Figure 4A, during the multiple F-T processes, the water-holding ability of the shrimp tended to decrease and was lower than that of the cold-storage group. This was mainly due to the repeated formation of ice crystals in *Penaeus vannamei* during repeated freezing and thawing, and the size, shape, and position were different. Severe damage to cells and cell membranes and protein freezing denaturation weaken the water-binding capacity and decrease the muscle water-holding capacity [50]. After seven F-T cycles, the water-holding power of shrimp meat increased, possibly because the repeated F-T cycles damaged the shrimp cell structure and caused water loss during thawing. When the water-holding capacity was calculated, the collected water was less than the actual outflow, so the increase in the water-holding power of shrimp meat was an illusion.

#### 3.5.2. LF-NMR and MRI

Flavor changes and protein denaturation in seafood are related to the state of water molecules. Through LF-NMR, the quality of products can be evaluated by detecting water distribution and water migration. Appendix A shows the change in relaxation time T_2_ in the cyclic F-T group. Four peaks of different sizes were observed for each relaxation time. The relaxation time range of 0.1–10 ms is the bound water (T_21_), which usually refers to the water attached to molecular bonds of components such as proteins; T_22_ does not easily allow water to flow, and the relaxation time is within 10–100 ms, which is mainly located in the water between myofibrils and other tissues. The water state easily lost in the shrimp is free water (T_23_), and the relaxation time is within 100–1000 ms. With a shorter relaxation time, the combination of material and water becomes closer, and the water is difficult to lose. Therefore, as the number of F-T cycles increased, the T_2_ values shifted to the right, indicating that the fluidity of water molecules in shrimp was enhanced. Figure 4B shows that the signal intensity and peak area of bound water and immobile water remained stable, while the relaxation time of free water increased continuously, and the peak area decreased from the initial 2.34% to 0.83%. This indicated that under repeated F-T, the size and position of ice crystals were different, which destroyed the muscle tissue, myofibrillar, and protein grid structure, resulting in a decrease in the water retention capacity of muscle tissue.

The brightness change in the pseudocolor map of nuclear magnetic resonance imaging can directly reveal the change in water content and distribution in shrimp meat. In general, when the moisture content in the sample is higher, the color in the pseudocolor image tends to be red; on the contrary, it tended to be blue. As seen from Figure 4B, with increasing storage time, the color of the pseudocolor image of the shrimp gradually becomes the background color, indicating that the water is slowly discharged from the inside of the myofibril to the gap of the myofibril, and then flows to the surface of the shrimp in the form of drip [17]. In addition, at the same storage time, the pseudocolor map of the F-T group changed significantly, resulting in more severe water migration and loss, consistent with the water content results.

### 3.6. Analysis of Shrimp Freshness Model

#### 3.6.1. The Freshness of Different Odors

The electronic nose was used to collect the odor of shrimp samples stored under different conditions for PCA modeling and to analyze the aroma changes due to freshness variations. As shown in Figure 5, the freshness level of shrimp under other storage conditions can be well distinguished. The sum of the first and second principal components is more significant than 99%, indicating that it can effectively reflect the information in the data. Among them, the cold-storage group can better distinguish the samples with different freshness, while the volatile substances produced in the late storage of shrimp in the F-T cycle group are the same and cannot be distinguished. Therefore, as the number of freeze–thaw cycles increases, the spoilage odor produced in the body gradually increases, and the emission becomes more apparent. LDA reached the maximum variation within and between PCA groups [51]. The results showed that the total contribution rate of the cold-storage and F-T cycle groups was over 82.3%, and the freshness could be separated. The data point clustering effect was good, and the data were relatively reliable, indicating that the volatile substances of shrimp in the freeze–thaw cycle process showed a stepwise change. According to the results of the loading analysis, W1S, W1W, W2S, and W5S played an essential role in the discrimination of the different sensory freshness levels of *Penaeus vannamei*. This is mainly because, during storage, microorganisms’ continuous growth and reproduction further decompose the shrimp muscle protein to produce trimethylamine, ammonia, hydrogen sulfide, and other substances, gradually emitting a rotten odor.

#### 3.6.2. Sensor Discrimination Model for Different TPC Contents

Appendix A shows the discriminant model analysis of shrimp with different TPC contents. From the PCA and LDA, it can be seen that the odor characteristics of TPC were similar when it transitioned from 4–5 lg CFU/g to 5–6 lg CFU/g; therefore, the substances could not be accurately differentiated. However, when the TPC was higher than 6 lg CFU/g, the rapid growth and reproduction of microorganisms made the volatile substances gradually increase, and the emission of odor was evident, which was easy to distinguish from the first-grade freshness and the second-grade freshness. The loading analysis showed that the W1W, W1S, W2S, and W5S sensors played an essential role in judging the TPC and the sensory freshness, and the combination of the different sensory freshness loadings. The analysis showed that the increase in TPC with the rise in the number of F-T cycles was mainly due to the production of sulfide-like substances.

#### 3.6.3. Sensor Discrimination Model for Different TVB-N Contents

Appendix A is the discriminant model analysis diagram of different TVB-N contents. PCA and LDA distinguished the data point set of >30 mg/100 g, but the transition range between <20 mg/100 g and 20–30 mg/100 g was challenging to distinguish. Compared to the PCA model, the model established using LDA was better. Combined with the analysis of different TPC discriminant models, the loading analysis showed that TVB-N produced during cyclic F-T was consistent with TPC, which may be related to the formation of volatile base nitrogen during the growth and reproduction of microorganisms. In general, the W1W, W1S, W2S, and W5S sensors were related to the quality changes of *Penaeus vannamei* and could be used as the primary sensors to detect the freshness of *Penaeus vannamei* to realize the rapid and nondestructive flavor detection of an electronic nose.

#### 3.6.4. Correlation Analysis

Figure 6 is the correlation heatmap of the sensory evaluation, TPC, TVB-N, and electronic nose sensor response value of shrimp under different storage conditions. As shown in the figure, the storage time of shrimp was significantly positively correlated with TVB-N and TPC (r > 0.95), but significantly negatively correlated with sensory evaluation (r < −0.96), indicating that the changes in TPC, TVB-N, and sensory assessment caused the decrease in the freshness of shrimp during storage. Sensory evaluation was highly negatively correlated with TPC and TVB-N (r < −0.88). The increase in storage time could lead to a decreased sensory evaluation by affecting the rise of the TVB-N and TPC content. The sensory assessment had a moderate negative correlation (r < 0.59) with W5S, W1S, W1W, W2S, and W2W. TPC and TVB-N were moderately positively correlated with W5S, W1S, W1W, W2S, and W2W (r > 0.58), indicating that the W5S, W1S, W1W, W2S, and W2W sensors were sensitive to the odor characteristics of shrimp during storage and could be used as the primary sensors to judge the freshness of *Penaeus vannamei*.

## 4. Conclusions

In this experiment, prepackaged *Penaeus vannamei* meat was examined to investigate changes in physicochemical properties under different storage conditions. The shift in freshness of shrimp was analyzed after cyclic freezing and thawing by combining it with electronic nose technology. The freshness test showed that the freshness of shrimp started to decline significantly after 8 F-T cycles, and 20 F-T cycles was the maximum number of cycles acceptable for freshness. The LF-NMR results showed that the T_2_ value of shrimp gradually migrated to the right, the mobility of water molecules was enhanced, and the transverse relaxation time of free water increased. According to the electronic nose correlation analysis, there was a high correlation between the W5S, W1S, W1W, W2S, and W2W sensors and storage time, sensory evaluation, TPC, and TVB-N, which meant that these were the primary sensors for judging the freshness of prawns and could be used to establish a model for quickly discriminating the freshness of *Penaeus vannamei*. This study provides an experimental basis for the quality control of *Penaeus vannamei* during marketing and storage, and can further provide a reference for aquatic products’ cold chain freshness research.

## Figures and Tables

**Figure 1 foods-13-00305-f001:**
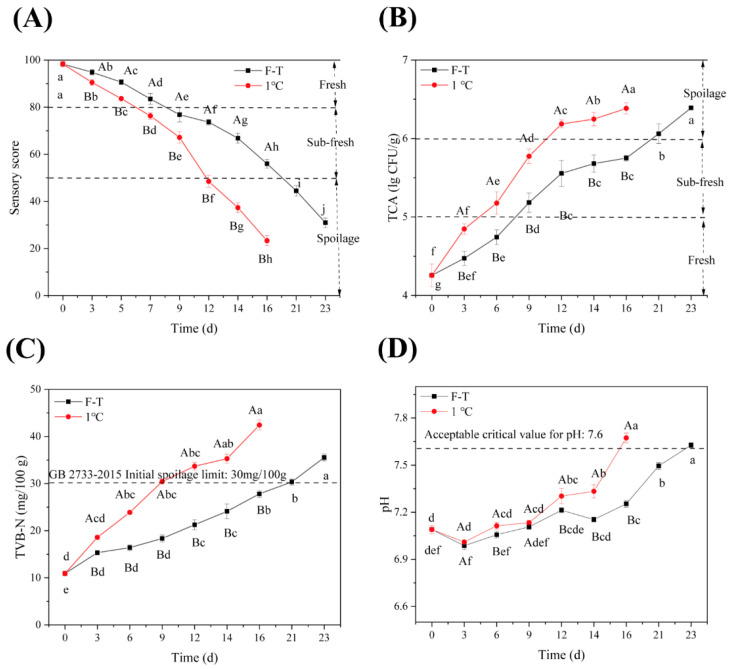
Physicochemical properties of *Penaeus vannamei* under different storage conditions. (**A**) Sensory evaluation; (**B**) TPC; (**C**) TVB-N; (**D**) pH. (The F-T cycle is set as a cycle of −20 °C storage for 12 h followed by 1 °C storage for 12 h). Different capital letters indicate significant differences between different storage methods for the same storage time. Different lowercase letters indicate significant differences between different storage times for the same storage method (*p* < 0.05).

**Figure 2 foods-13-00305-f002:**
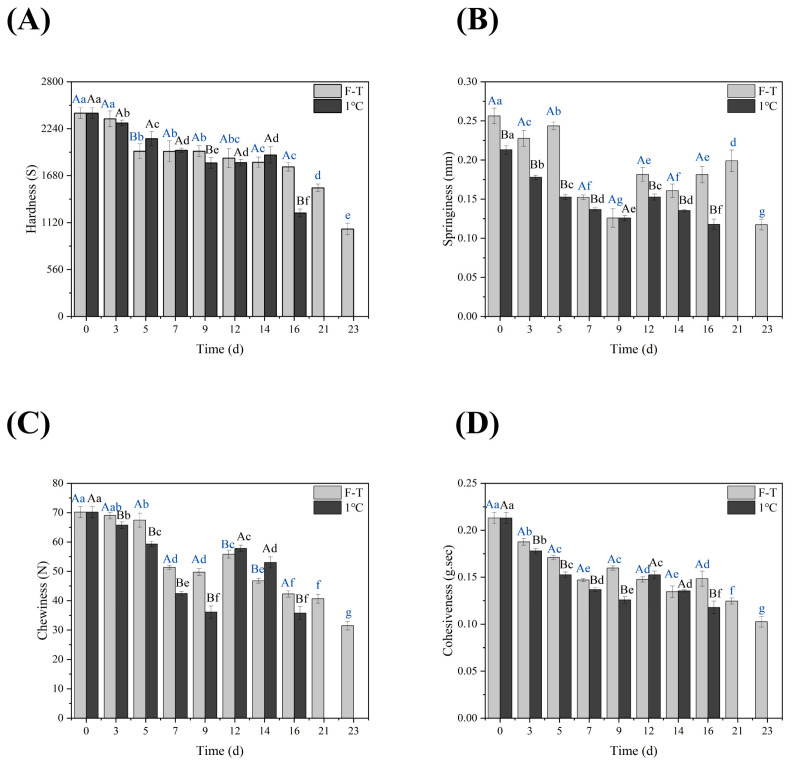
Textural characteristics of shrimp meat under different storage conditions. (The F-T cycle is set as a cycle of −20 °C storage for 12 h followed by 1 °C storage for 12 h). Different capital letters indicate significant differences between different storage methods for the same storage time. Different lowercase letters indicate significant differences between different storage times for the same storage method (*p* < 0.05). (**A**) Hardness; (**B**) Springiness; (**C**) Chewiness; (**D**) Cohesiveness.

**Figure 3 foods-13-00305-f003:**
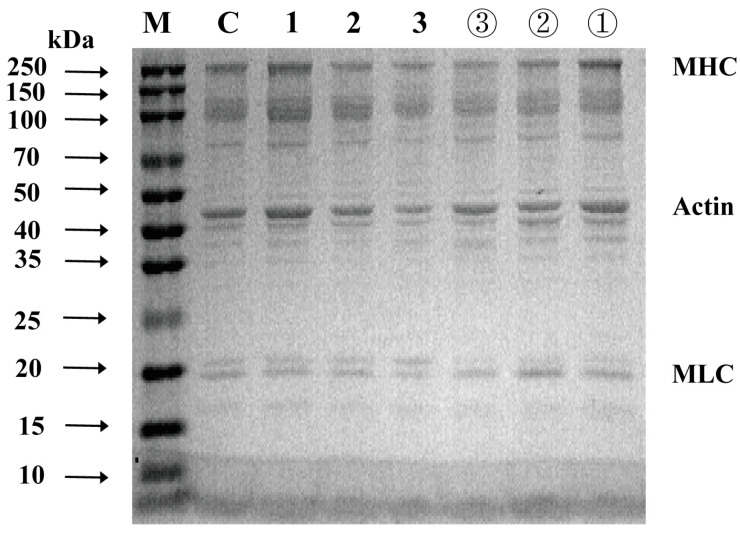
SDS-PAGE profiles of myofibrillar proteins from shrimp with different freshness levels (M: marker, C: control group, 1, 2 and 3: fresh, sub-fresh, and spoilage of F-T group, ①, ②, ③: fresh, sub-fresh, and spoilage of cold-storage group, MHC: myosin heavy chain, MLC: myosin light chain). (The F-T cycle is set as a cycle of −20 °C storage for 12 h followed by 1 °C storage for 12 h).

**Figure 4 foods-13-00305-f004:**
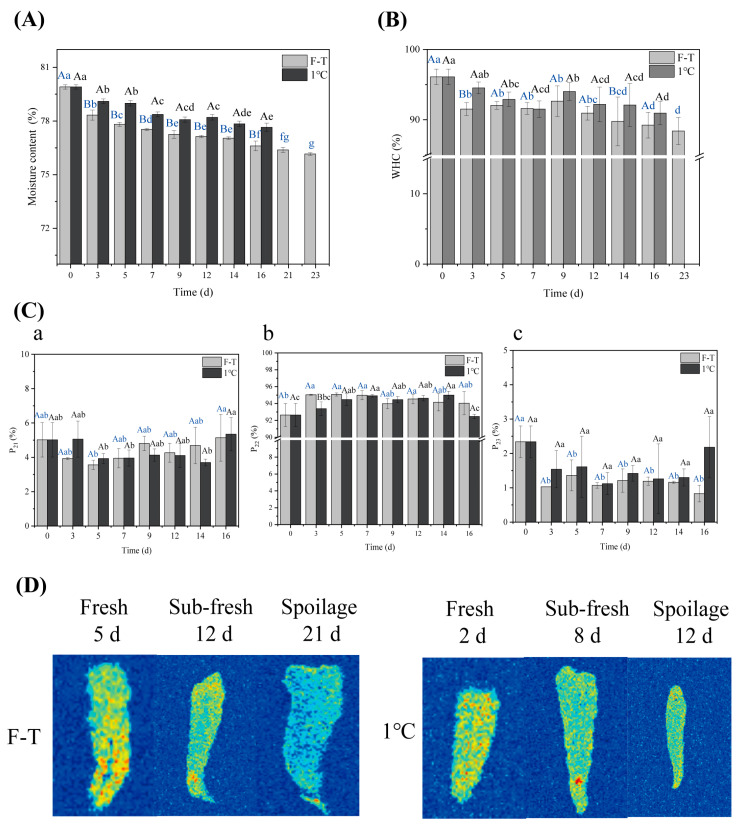
Moisture changes in shrimp meat: (**A**) Moisture content, (**B**) WHC; (**C**) Proportion of peak area in T_2_ relaxation time of shrimp: (**a**: T_21_, **b**: T_22_, **c**: T_23_); and (**D**) ^1^H-MRI pseudocolor maps. (The F-T cycle is set as a cycle of −20 °C storage for 12 h followed by 1 °C storage for 12 h). Different capital letters indicate significant differences between different storage methods for the same storage time. Different lowercase letters indicate significant differences between different storage times for the same storage method (*p* < 0.05).

**Figure 5 foods-13-00305-f005:**
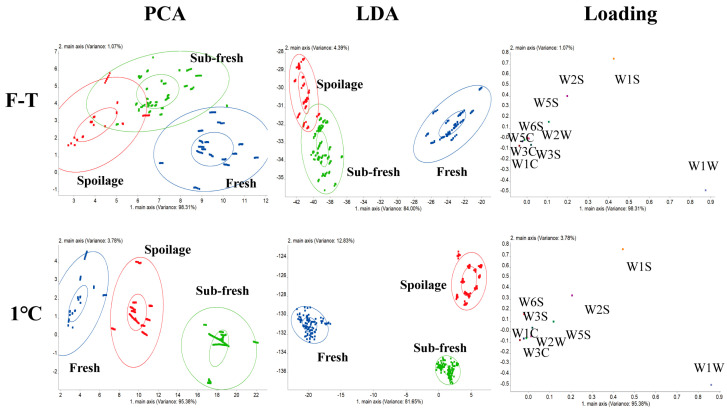
Plot of PCA, LDA, and loading analyses for different sensory freshness levels of shrimp. (The F-T cycle is set as a cycle of −20 °C storage for 12 h followed by 1 °C storage for 12 h).

**Figure 6 foods-13-00305-f006:**
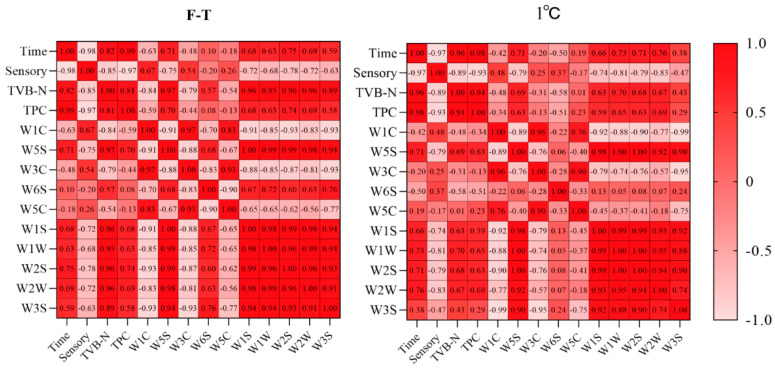
Correlation analysis of shrimp sensory evaluation, TPC, and TVB-N with sensor response values. (The F-T cycle is set as a cycle of −20 °C storage for 12 h followed by 1 °C storage for 12 h). Note: Darker color indicates a stronger positive correlation; lighter color indicates a stronger negative correlation.

## Data Availability

The original contributions presented in the study are included in the article/Appendix A, further inquiries can be directed to the corresponding author.

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
