# Peer review of "Effect of Freeze–Thaw Cycles on the Freshness of Prepackaged Penaeus vannamei"

_foods, 2024, doi:10.3390/foods13020305_

Round 1

Reviewer 1 Report

Comments and Suggestions for Authors

To the esteemed Authors:

The manuscript is written well and the different sections of the article are set correctly. I think the manuscript will be useful for the readers. But some corrections in the manuscript is need for enhance the quality of the report. First, you don’t have data about shrimp maintaining in the freeze temperature without defrost cycle. It is important that readers know if the cold chain keep correctly without fluctuant the quality assessments have significant difference by your thawing cycle data? Second things that must be correct is the statistical analysis inside figures and in data reports. These corrections are set in the below comments.

1-     The authors just compare the (F-T) treatment by 1 °C. I think it is important to know if the temperature was steady state at -20 the quality assessments showed any significant difference by other two treatments? This section doesn’t analyze by the authors.

2-  SDS-PAGE section figures don’t support the discussions inside the manuscript and must be completely revised.

3-     statistical analysis inside figures and in data reports need major corrections.

4-     The comparison by other works is rare in the discussions.

The detail of some corrections or errors are listed below:

Line 25: Penaeus vannamei must be in italic mode.

Line 20-27: rewrite and rearrange it. first bring all results and finally conclude them.

Keywords: eliminate "Shelled" and "Correlation" and add keywords that could reflect the aim of the research.

Line 42: low-temperature or freeze-temperature?

Line 46-50: It is better to bring the standard temperature for storage and transportations of the shrimp in China and other international standards!

Line 53: it is better to add "...many ice crystals that most of them is big in size and growth in every next F-T cycle. (Reference?)" to final the sentence.

Line 95: replace "shrimp line" by a better word.

Line 97-100: You must set a negative control (-20-degree centigrade in the whole period) to compare with the treatments and positive control.

line103: add the age range of the evaluators.

Table S1: correct the scientific name: Litopenaeus vannameini?

Line 112: how much diluted sample added to the petri dish?

Line 128: explain briefly how you separate and analyzed the second abdominal segment?

Line 150-159: Do your instrument measure moisture content automatically or you calculate it?

Line 156: mentioned if you add water at the first before centrifugation?

Line 160: rewrite formula by Math type software.

line 196: you don’t assay oxidation so it is better to substitute " mainly" by "may be".

Line 202: please explain how the 20 cycles is calculated?

Line 211: 1(c) or 1(D)?

Line 213: add "may be" to the sentence. you didn’t analyze the glycogen!

Line 236: 1(D) or 1(C)??

Figure 2: add differential statistical letters between two different groups on the columns

Figure 2 (B)(C)(D): after day 9 or 7 you have some increase in the measures! please discuss about these enhancements and add references

Line 269-275: the discussion of result not acceptable. based on the figure 3, control has different bands by the fresh samples of two treatments, Also MHC and actin is sharper and tick in sub fresh and spoilage treatments. rewrite this section again or bring a sharper SDS-PAGE picture

Figure 3: explain MLC, PM, MHC in the legend

Line 285-287: Add reference

Line 306: add reference

Figure 4: add differential statistical letters between two different groups on the columns

Figure 4 (B): exit from bold style, also explain the T23, T2 and T21 in the legend

Line 327-334: you must change the reporting style of the figure 4(B), separate T21, T2 and T23 and bring each of the two treatments beside together and compare statistically. This style of the reporting is more scientific and the reader could completely understand the changes between two different treatments

Conclusion:

this is not acceptable because the results repeated again. rewrite the conclusion in shorter paragraph and avoid of bring results again.

Comments on the Quality of English Language

The manuscript needs minor English editing.

Reviewer 2 Report

Comments and Suggestions for Authors

Add high quality pictures of the samples and experimental setup.

All Figures: Add the temperatures and times of the F-T circles in the captions. Pictures should be readable without the main document.

Add more literature about freezer burn and typical temperature fluctuations during freeze storage.

L. 14-27: Add temperatures of the circles.

L. 38-40: How low is the shelf life then and against what deterioration these help?

L. 43: Be more precise. At what temperature enzymes lose activity, is it known? And when microbial growth is stopped?

L. 45: Normally quick freezing avoids the formation of ice crystals. How quick is the freezing?

L. 45: What temperature is “low-temperature”?

L. 47: What temperatures?

L. 51-55: Are values available? Scientific studies should be quantitative.

L. 58-60: Too general.

L. 60-72: Try to add values. What are the temperatures?

L. 31-81: Missing: Water vapour pressure at different temperatures and freezer burn.; What temperature fluctuations occur, refrigerators were optimised recent years.

L. 84-91: Product number is missing. Did Penaeus vannamei was treated e.g. salt?

L. 97-98: These conditions seem drastic. Are these realistic?

Fig. 1: Are there values available for -20 °C? That would be a reference. Could error bars be added?

Figure 4: Caption is too short to understand the Figures. Add more relevant information.

Round 2

Reviewer 1 Report

Comments and Suggestions for Authors

The reply to the answers is acceptable.

Author Response

Thank you for your suggestions, all of your suggestions are very important and they have guided me in writing my dissertation and in my research work!

Reviewer 2 Report

Comments and Suggestions for Authors

Thank you for ammendments!

Still, a high quality fotograph of the packaged product would be of values.

Much discussion is in the response letter: Please choose open review, so others can read this also.

Authors should look for older references, e.g. books about shelf life that are 20 to 30 yrs. old. In the past research was done on this topic.
